# Rainfall interception and redistribution by a common North American understory and pasture forb, *Eupatorium capillifolium* (Lam. dogfennel)

D. Alex R. Gordon[1,6], Miriam Coenders-Gerrits[2], Brent A. Sellers[3,4], S.M. Moein Sadeghi[5], John T. Van Stan II[1,6]

[1] Geology & Geography, Georgia Southern University, Statesboro, GA, USA

[2] Delft University of Technology, Water Resources Section, Stevinweg 1, 2628 CN Delft, The Netherlands

[3] Agronomy, University of Florida, Gainesville, FL, USA

[4] Range Cattle Research & Education Center, University of Florida, Institute of Food & Agricultural Sci, FL, USA

[5] Department of Forestry & Forest Economics, University of Tehran, Karaj, Iran

[6] Applied Coastal Research Lab, Georgia Southern University, Savannah, GA, USA

*Correspondence to*: Miriam Coenders (A.M.J.Coenders@tudelft.nl)

**Abstract.** In vegetated landscapes, rain must pass through plant canopies and litter to enter soils. As a result, some rainwater is returned to the atmosphere (i.e., interception, $I$) and the remainder is partitioned into a canopy (and gap) drip flux (i.e., throughfall) or drained down the stem (i.e., stemflow). Current theoretical and numerical modelling frameworks for this process are near-exclusively based on data from woody overstory plants. However, herbaceous plants often populate the understory and are the primary cover for important ecosystems (e.g., grasslands and croplands). This study investigates how overstory throughfall ($P_{T,o}$) is partitioned into understory $I$, throughfall ($P_T$) and stemflow ($P_S$) by a dominant forb in disturbed urban forests (as well as grass- and pasturelands), *Eupatorium capillifolium* (Lam., dogfennel). Dogfennel density at the site was 56,770 stems ha$^{-1}$, enabling water storage capacities for leaves and stems of 0.90±0.04 mm and 0.43±0.02 mm, respectively. As direct measurement of $P_{T,o}$ (via tipping buckets or bottles, etc.) would remove $P_{T,o}$ or disturb the understory partitioning of $P_{T,o}$, overstory throughfall was modelled ($P'_{T,o}$) using on-site observations of $P_{T,o}$ from a previous field campaign. Relying on modelled $P'_{T,o}$, rather than on observations of $P_{T,o}$ directly above individual plants, leaves significant uncertainty regarding (i) small-scale relative values of $P_T$ and $P_S$ and (ii) factors driving $P_S$ variability among individual dogfennels. Indeed, $P_S$ data from individual plants were highly skewed, where mean $P_S$:$P'_{T,o}$ per plant was 36.8%, but the median was 7.6% (2.8%-27.2% interquartile range) and total over the study period was 7.9%. $P_S$ variability ($n = 30$ plants) was high ($CV > 200\%$) and may hypothetically be explained by fine-scale spatiotemporal patterns in actual overstory throughfall (since no plant structural factors explained the variability). Total $P_T$:$P'_{T,o}$ was 71% (median $P_T$:$P'_{T,o}$ per gauge was 72%, 59-91% interquartile range). Occult precipitation (mixed dew/light rain events) occurred during the study period, revealing that dogfennel can capture and drain dew to their stem base as $P_S$. Dew-induced $P_S$ may help explain dogfennel's improved invasion efficacy during droughts (as it tends to be one of the most problematic weeds in the southeastern US's improved grazing systems). Overall, dogfennel's precipitation partitioning differed markedly from the site's overstory trees (*Pinus palustris*), and a discussion of the limited literature suggests that these differences may exist across vegetated ecosystems. Thus, more research on herbaceous plant canopy interactions with precipitation is merited.

**Key words**: Rain, throughfall, stemflow, canopy water storage, stem water storage, evaporation.

## 1. Introduction

Precipitation ($P_g$) across most of the global land surface will interact with plant canopies. Precipitation-canopy interactions during storms result in three general hydrologic processes; one which returns water to the atmosphere (interception) and two others that route water to the surface (throughfall and stemflow). Interception is the evaporation of droplets splashing against (Dunkerley, 2009), or stored on, canopy surfaces, like leaves (Pereira et al., 2016), bark (Van Stan et al., 2017a, and epiphytes (Porada et al., 2018). Depending on the vegetation and storm conditions, interception can be small per unit area (David et al., 2006) or return half the annual precipitation to the atmosphere (Alavi et al., 2001). In this way, canopy interception can evaporatively cool regions (Davies-Barnard et al., 2014), recycle moisture to generate nearby storms (Van der Ent et al., 2014), and reduce stormwater runoff to save millions of dollars (US) in stormwater infrastructure costs (Nowak et al., 2020). Throughfall is the water that drips to the surface through gaps or from canopy surfaces, while stemflow is the water that drains down plant stems. The portion of precipitation that drains as throughfall versus stemflow is also highly variable depending on vegetation and storm conditions: ranging annually from 10-90% for throughfall and <1-60% for stemflow (Sadeghi et al., 2020). Since throughfall and stemflow reach the surface at different locations, they differentially interact with subsurface hydrological and biogeochemical processes—having been implicated in fine-scale patterns in soil physicochemistry (Gersper and Holowaychuk, 1971), microbial community composition (Rosier et al., 2015; 2016), N-cycling functional genes (Moore et al., 2016), and metazoan community composition (Ptatscheck et al., 2018). Accurate accounting for each of these precipitation partitioning fluxes is, therefore, necessary for the accurate prediction of atmospheric and surface hydro-biogeochemical processes.

Current theoretical and numerical modeling frameworks for canopy precipitation partitioning (see review by Muzylo et al. (2009)), are almost exclusively based on observations beneath woody plants, like forests and shrublands (Sadeghi et al., 2020). In forests, the past 150 years of research has primarily targeted dominant overstory trees (Ebermayer, 1873; Van Stan and Gordon, 2018). However, herbaceous plants commonly dominate forest understories and can be abundant beneath shrublands (Jiménez-Rodríguez et al., 2020; Lajtha and Schlesinger, 1986; Specht and Moll, 1983). As a result, our current understanding of "net" precipitation (as measured beneath woody overstory canopies) is not representative of the actual precipitation that reaches the surface (or litter layer: Gerrits and Savenije, 2011) beneath the understory. Herbaceous canopies are relevant to precipitation partitioning in more than the one-third of the global land surface represented by forests; they also cover 27% and 11% of the global land surface in grasslands and croplands, respectively (Alexandratos and Bruinsma, 2012; Suttie et al., 2005). It is unlikely that current knowledge on precipitation partitioning based on woody vegetation is applicable to herbaceous vegetation, since they differ in many hydrologically-relevant morphological features: smaller height, the lack of bark structure, and presence of other stem features (like trichome hairs or desiccated leaves), etc. This raises unanswered and little-researched, questions that must be addressed to include herbaceous plants in precipitation partitioning theory, e.g.: How do these significant morphological differences affect canopy and stem water storage capacities? Do herbaceous plants also favor throughfall generation, like woody plants, or do they more efficiently drain precipitation to their stem bases (and, thereafter, their shallow roots)? In fact, several long-standing (and hitherto unanswered) calls for greater research on the precipitation partitioning of non-woody plants (rooted in detailed observations) have been made (Price

et al., 1997; Price and Watters, 1989; Verry and Timmons, 1977; Yarie, 1980). These are general questions identified
by the community; but, in this study we focus on: How is overstory throughfall ($P_{T,o}$: Figure 1) partitioned into
understory interception, throughfall ($P_T$: Figure 1) and stemflow ($P_S$: Figure 1) by a dominant forb in disturbed urban
forest understories (as well as grass- and pasturelands), *Eupatorium capillifolium* (Lam., dogfennel)?
Very little is known about how understory plants partition $P_{T,o}$ into understory $P_T$ and $P_S$ (Figure 1). Overstory
stemflow is currently assumed to bypass the understory and litter layers (Carlyle-Moses et al., 2018); however, this
assumption, particularly regarding the bypass of litter, has rarely been tested (Friesen, 2020) and overstory stemflow
has been observed to runoff for long distances away from the stem (Cattan et al., 2009; Keen et al., 2010). We do not
investigate interactions between the understory and overstory stemflow in this study, because stemflow from this study
site is negligible (<0.2%: Yankine et al., 2017). Most observations of precipitation partitioning beneath any plant
besides overstory woody plants have been done on maize (Zheng et al. (2019) and references therein) and other cash
crops (Drastig et al. (2019) and references therein), which leave plants of forest understories, grasslands or
pasturelands relatively unresearched. Even the few studies on forest understory interception, $P_T$, and $P_S$
overwhelmingly focus, again, on woody plants (González-Martínez et al., 2017; Price and Watters, 1989), limiting
net precipitation observations beneath understory herbaceous plants to ferns (Verry and Timmons, 1977) and
nonvascular plants (Price et al., 1997). These scant observations, however, indicate that precipitation partitioning by
non-woody understory plants is hydrologically relevant, as they can store as much water as woody plants (Klamerus-
Iwan et al., 2020), evaporate significant portions of $P_{T,o}$ (Coenders-Gerrits et al., 2020) and redistribute 7-90% of event
$P_{T,o}$ as $P_S$ (Sadeghi et al., 2020). For our study on dogfennel, we hypothesized that, compared to past research on
woody plants, dogfennel stems and leaves (i) can store a hydrologically relevant amount of rainwater (i.e., within the
range of water storage capacities reported for woody plants: (Klamerus-Iwan et al., 2020), (ii) significantly reduce net
rainfall flux to the surface (i.e., $P_T + P_S << P_{T,o}$), and (iii) redistribute a substantial portion of $P_{T,o}$ to the surface via $P_S$
(i.e., $P_S$ will often "funnel" more rainwater per storm to the soils surrounding stems than $P_T$, $P_{T,o}$ or $P_g$ over the same
area). To test these hypotheses, $P_{T,o}$ was modelled from past on-site observations ($P'_{T,o}$) as monitoring $P_{T,o}$, $P_S$, and $P_T$
simultaneously were not possible without disrupting or removing $P_{T,o}$. We explicitly acknowledge that the decision to
rely on modelled $P'_{T,o}$ leaves a non-trivial uncertainty regarding the influence of actual overstory throughfall
spatiotemporal patterns on small-scale values of $P_T$ and individual plants' $P_S$.

## 2. Materials and methods

### 2.1. Study site and study plant description

The study site, Herty Pines, is a forest fragment in Statesboro, Georgia, USA (Figure 2a), at Georgia Southern
University's main campus (32.430 N, -81.784 W, 65 m A.S.L.). Climate is subtropical (Köppen *Cfa*) where mean
monthly temperatures (1925-2014) for July range from 21-33°C and winter months are generally mild, i.e., the lowest
mean January temperature is 3.5°C (University of Georgia, 2019). Mean annual precipitation is 1,170 mm y$^{-1}$ and
precipitation occurs almost exclusively as rain, relatively evenly spread over the year. The overstory is dominated by
*Pinus palustris* (longleaf pine) and overstory rainfall partitioning for this site has been reported (Mesta et al., 2017;
Van Stan et al., 2018; Yankine et al., 2017). Trunk diameter at breast height (DBH) was relatively consistent across
all trees in the study plot, 49.7 cm (mean) with an interquartile range of 36.2-55.7 cm. Mean tree height was 30.4±4.5
m and was derived from terrestrial lidar (terrestrial lidar methods identical to Van Stan et al., 2017a). Stand density
was 223 trees ha$^{-1}$ with 50.4 m$^2$ ha$^{-1}$ of basal area. Dogfennel, our study plant, was particularly dominant along the
forest edge. Dogfennel is a forb of the Asteraceae family, native to (and widespread across) North America (Van
Deelen, 1991; Wunderlin and Hansen, 2003). Although dogfennel behaves as an annual plant throughout much of its
North American range, it can behave as a perennial in the southern US by overwintering as a rosette, typically from
January to March, before re-growing from a taproot in the spring, typically in April (Macdonald et al., 1994;
Macdonald et al., 1992). Dogfennel can be abundant in disturbed forest understories, particularly pine forests
(Brockway et al., 1998) and pastures (Figure 2b). In the study pine forest, dogfennel stem density was 56,770 stems
ha$^{-1}$ along the stand edge. In pasturelands, dogfennel can reach this stem density within a single season and, if left
unmanaged, dogfennel densities have been measured as high as 74 stems m$^{-2}$, or ~740,000 stems ha$^{-1}$ (Dias et al.,
2018). The growth habit of dogfennel results in "clumps" of stems. Dogfennel density was estimated in ten 10x10 m
plots by counting the stems clump$^{-1}$ for 3 randomly-selected clumps in each plot. For each plot, the mean stems clump$^{-1}$
were multiplied by the number of clumps plot$^{-1}$. Finally, all stems plot$^{-1}$ were summed and scaled to 1 ha. Three
dogfennel clumps were randomly selected for throughfall and stemflow monitoring. Within these three clumps, 30
individual dogfennel stems were randomly selected for stemflow monitoring. Individual plant attributes—canopy
radius [cm], stem radius [cm], leaf angle at the stem [degrees from vertical] at various canopy heights (1.00, 1.25,
1.50, 1.75, 2.00 m), and relative location within the clump, interior (I), middle (M), or exterior (E)—were measured
for each stemflow-instrumented plant (Table 1). Canopy and stem radii were determined manually with a tape
measure, where canopy radii were the mean of measurements from eight directions (N, NE, E, SE, S, SW, W, and
NW) and stem radius was determined by a single manual measurement at the stem base. Leaf angle at the stem was
determined for two leaves at each height using the iProtactor App for iPhone (2013, Phoenix Solutions) which logs
an angle after the levelling of the iPhone camera (see Figure S1 for example).
**2.2. Hydrometeorological monitoring**
**2.2.1. Rainfall measurements**
Rainfall amount, duration and intensity for discrete rain events were automatically logged every 5 min by a weather
station installed above the canopy (on the rooftop of nearby Brannen Hall at ~40 m height), which is located 100 m
from Herty Pines. Rainfall observations were recorded by three tipping bucket gauges (TE-525MM, Texas
Electronics, Dallas, TX, USA) interfaced with a CR1000 datalogger (Campbell Scientific, Logan, Utah, USA). This
weather station logged a suite of other meteorological variables; however, since these data do not represent the
meteorological conditions experienced by the understory, they are not reported or examined here. A discrete event
was defined as any atmospheric moisture (rainfall or dew) that resulted in a measurable quantity of throughfall and
stemflow (more than a few mL) that occurred after a minimum interstorm dry period of 8 h. Few events consisted of
early morning dew contributions (visually observed during sampling and verified by air temperatures equalling dew
point temperatures), and these occurred after low-magnitude nighttime rainfall. When dew was present in the
understory, there was no response from above-canopy rain gauges; thus, a post-hoc estimate of occult dew contribution
to $P_{T,o}$ was made by assuming the dew contribution was equal to the understory canopy water storage capacity (1.33
mm – methods described later). An important limitation to this dew estimate is that it represents the maximum possible
dew contribution. Rain events without dewfall required at least ~4 mm of rainfall for generation of $P_T$ or $P_S$ from the
monitored dogfennel canopies.

**2.2.2. Overstory throughfall estimation**

As observing $P_{T,o}$ directly would prevent direct observation of $P_T$ and $P_S$ beneath dogfennel plants, $P_{T,o}$ was estimated
from previous field measurements at the site (Figure 3). Automated $P_{T,o}$ monitoring was performed from September
2016 to September 2017 using ten 3.048-m long and 10.16 cm diameter PVC troughs oriented at a moderate slope,
with a 5.08 cm slot cut lengthwise for collection and drainage of $P_{T,o}$ to a Texas Electronics (Dallas, Texas, USA) TR-
525I tipping bucket gauge, resulting in 1.65 $m^2$ of collection area. Tipping bucket gauges and their associated troughs
were randomly placed within a 0.25 ha plot and recorded every 5 minutes by a CR1000 datalogger. All trough angles
were measured with a digital clinometer to correct computations of trough area receiving $P_{T,o}$. Trough and tipping
bucket assemblies were field tested to ensure accuracy (± 5%) under storm conditions typical for the region (Van Stan
et al., 2016). These $P_{T,o}$ data were reported by Mesta et al. (2017). To estimate overstory throughfall, $P'_{T,o}$, a regression
model was generated from the association between $P_{T,o}$ [% of rainfall] measured on site and storm size, $R$ [mm storm-
$^1$] using the "Aston" curve:
(2) $P'_{T,o} = a\,(1 - e^{-b\,R})$
where $a$ and $b$ are regression coefficients. This model and its fit statistics are provided in Figure 3. We assume that
the past observed rainfall relationship with $P_{T,o}$ at the site was similar during our study period. Although we are unable
to assess whether and to what degree there is a difference between these observation periods, the canopy is mature
and there has been no known/noticeable disturbance or change in canopy structure since the previous observation
period.

**2.2.3. Understory throughfall and stemflow measurements.**

Throughfall gauges consisted of 9 randomly placed funnels (506.7 $cm^2$ collection area each), three per
dogfennel clump (1,520.1 $cm^2$ total collection area per clump), connected to HDPE bottles that were manually
measured with graduated cylinders immediately after a storm ended (within 4 h). The total canopy area of dogfennel
plants at this site rarely exceed 2,000 $cm^2$, resulting that the total throughfall gauge area per clump generally
represented >75% of canopy area; which is a comparatively much larger gauge-to-canopy area than most past
throughfall studies on forest canopies (Van Stan et al., 2020).
Standard stemflow measurement methods developed for woody plants (use of flexible tubing wrapped around
a woody stem: Sadeghi et al., 2020) are not suitable for dogfennel; moreover, no standard stemflow collection devices
exist for herbaceous plants. Thus, stemflow collars were constructed from aluminum foil, 15-mm inner-diameter
flexible polyethylene tubing, electrical tape, and silicon (see Figure S2). Aluminum foil was folded over itself several
times to strengthen the collar (typically ~160 mm length of foil was folded to ~40 mm) and connected to plastic tubing
with stainless steel staples. The aluminum collar was then folded around the lower stem of the dog fennel and secured
with electrical tape. To seal the aluminum foil, staple connections, and the interstices between the foil, tubing and
stem, silicon was thinned with hydrotreated light (95-100%) naphtha (VM&P Naphtha, Klean-Strip, Memphis TN
USA), allowing for it to completely fill the aluminum cone up to the tube opening and make a water-tight seal. While
naphtha-thinned silicon was poured into collars, the tube opening was covered. An additional benefit of naphtha-
thinned silicon was that, due to the evaporation of naphtha, the silicon shrinks, thereby, pulling the collar taut and
stiffening/strengthening the stemflow collection device and extending the lifespan of the collar. Stemflow was
measured with a graduated pipette (with 1 mL graduations) from 500 mL plastic bottles connected to the tubing base.

## 2.3. Water storage capacity estimation

Maximum water storage capacity ($S_u$ [mm]) was estimated for the dogfennel canopy and stem, both as volume [L] per
unit surface area [$m^2$]. All field leaf and stem samples were collected during an inter-storm dry period (>24 h after
any rainfall). For the canopy, 50 leaves representing the median size of the site dogfennel plants were sampled (broken-
off at the base of the leaf), taken back to the lab, their "field-dry" mass [g] determined on a bench scale, and then the
broken end of their leaf-stems were sealed with silicon to prevent water exchange from an area that was not previously
exposed in its natural state. Sampling for the stems was similar; however, since dogfennel heights reach (and can
exceed) 2 m, the stems were cut into 5 cm sections. Just as with the leaves, 50 representative samples of these stem
sections were weighed in the lab, then sealed with silicon on both ends. Next, all leaf samples and stem sections were
submerged in water for three days until achieving maximum saturation (per Van Stan et al., 2015), whereupon the
maximum saturation mass [g] was recorded. For comparison with the field-dry mass, all samples were oven-dried
until their mass no longer changed (mass recorded every 3 h), whereupon the oven-dried mass [g] was recorded. No
leaf or stem samples were oven dried longer than 15 h. The gravity convection oven (Isotemp, Fisher Scientific) was
set to 40 °C (confirmed with a standard thermometer). The maximum volume of all samples' water storage capacity
is the difference between saturation and oven-dried mass. The oven-dried leaves and stems did not visually appear to
be damaged (aside from the sampling cuts, obviously) and care was taken to ensure the plant samples were not
damaged. It is likely that internal (not externally intercepted) water was exchanged during this process; however, this
is not entirely problematic as plant surfaces are known to permit interaction between externally intercepted water and
internal water (Berry et al., 2019). Moreover, we explicitly acknowledge that although these submersion methods are
commonly used, they produce the "maximum" possible water storage capacity (hence, our objective to estimate
maximum water storage capacity), as multiple intrinsic and extrinsic factors of plant surfaces could reduce the
available water storage capacity in situ (Klamerus-Iwan et al., 2020).
Specific water storage capacity, $S_L$ [mL cm$^{-2}$], for the leaves and stems was determined by dividing the lab-
derived maximum volume [mL] by the samples' surface area [cm$^2$]. For leaves, after sampling, levelled photos of
each sample were taken on a grid system (every block representing 2.5 cm x 2.5 cm for scale), then the leaf images
were vectorized and processed for 2-D projected surface area using the "Measure Path" extension in Inkscape (v. 0.92,
Inkscape.org). An example vectorized image of leaf area is provided in the supplemental materials (Figure S3). Error
in this vector-based leaf surface area estimate was estimated by repeating the process five times for each leaf. Stem
surface area for all samples was estimated from their radii and height. $S_L$ estimates for the stem (0.436 mL cm$^{-2}$) and
leaves (0.195 mL cm$^{-2}$) were then scaled to $S_U$ [mm as L m$^{-2}$] using stem and leaf surface area estimates per plant ($A$
= 171.9 cm$^2$ plant$^{-1}$ and 807.5 cm$^2$ plant$^{-1}$, respectively), and multiplied by the site plant density ($D$ = 5.68 plants m$^{-2}$)
and divided by 1000:
(2) $S_U = (S_{L_{stem}} \times A_{stem} \times D)/1000 + (S_{L_{leaf}} \times A_{leaf} \times D)/1000$
Plant stem and leaf surface area estimates were determined from 5 representative plants that were cut from
the site and separated into leaves and stems, then the sum of leaf and stem areas (determined as mentioned earlier in
the paragraph) were divided by 5. Total leaf surface area compares well to values reported from ~1 m tall dogfennel
plants, 212 cm$^2$ plant$^{-1}$ (Carlisle et al., 1980), considering our plants were much taller (~2 m).
**2.4. Data analysis**
Descriptive statistics were compiled for all variables presented and regression analyses were performed to relate plant
canopy and hydrologic variables. All statistical analyses were done using Statistica 12 (StatSoft, Tulsa, OK, USA).
Throughfall volumes [L] from all gauges were summed and converted to yields [mm] by dividing by the total gauge
area [m$^2$]. Stemflow yield [mm] for an individual plant was determined by dividing its volume [L] by the projected
canopy area [m$^2$]. To compare stemflow production across plants, two metrics were computed per plant for each storm:
normalized stemflow ($\bar{P}_{S,i}$ [-]) and the funneling ratio ($F$ [-]). $\bar{P}_{S,i}$ was computed per Keim et al. (2005):
(3) $\bar{P}_{S,i} = \dfrac{\left(P_{S,i} - \bar{P}_S\right)}{s_S}$
where $P_{S,i}$ is stemflow volume [mL] from each individual plant in a single storm, $\bar{P}_S$ is the mean stemflow for all plants
in a single storm, and $s_S$ is the standard deviation of stemflow for all plants in a single storm. $F$ for individual plants
in each storm were computed per (Herwitz, 1986):
(4) $F = \dfrac{P_{S,i}}{B_i P}$
where $B_i$ is the basal area [cm$^2$] at the base of an individual plant and $P$ will be either $P_g$ or $P'_{T,o}$ (this will be explicitly
indicated in the results). There are an increasing number of $F$ metrics (Carlyle-Moses et al., 2018; Levia and Germer,
2015); however, the selected method is the most common $F$ metric applied to stemflow data to date. Moreover, in situ
observations of non-collared dogfennel plants during rainfall confirmed that dogfennel $P_S$ rates did not produce visible
runoff areas.
**3. Results**
**3.1. Storm and plant structural conditions**
Discrete rain events, as measured above the forest canopy, ranged in magnitude from 0.1 mm (during dewfall) to 101.3
mm (Table 1). The distribution of storm magnitudes was skewed, such that the mean, 16.5 mm, was many times
greater than the median, 6.6 mm (Table 1). Estimated overstory throughfall ($P'_{T,o}$), per Figure 3, ranged from 0 (again,
during dewfall) to 72.2 mm, with a median of 3.5 mm (Table 1). Thirty of the plants in the selected dogfennel clusters
- those being monitored for stemflow - had an average canopy radius of 18.3 cm (±4.5 cm standard deviation), which
was nearly identical to the median canopy radius (Table 1). The stem radii of all measured dogfennel plants ranged
from 0.1 - 0.7 cm, with a mean radius of 0.6 cm (Table 1). The resulting ratio of canopy:stem radii was also normally
distributed, with a mean and median of ~36 (dimensionless), but ranging from 24 to 50 (Table 1). For all plants, the
mean leaf angle decreased from 54° to 32° from vertical with increasing canopy height; i.e., the higher in the dogfennel
canopy, the closer the leaf angle approaches vertical (Table 1). This trend appears consistent across each individual
study plant regardless of which clump the plants' resided, as the standard deviation across all elevations are low, 1.8-
3.1° from vertical, and do not overlap (Table 1).

### 3.2. Partitioning into water storage, throughfall and stemflow

Note that $P'_{T,o}$ is an event-scale estimate derived from past observations, limiting its utility in examining fine-scale $P_T$
and individual-plant scale $P_S$. The sum of data from all storms throughout the study period resulted in $P_T$, $P_S$ and $I$ of
71%, 8%, and 21% as a portion of $P'_{T,o}$, respectively, beneath dogfennels at our site. Water storage capacity achieved
by dogfennel leaves in the lab was $0.90 \pm 0.04$ mm, while dogfennel stems stored a capacity of $0.43 \pm 0.02$ mm (Figure
4). This resulted in the total $S_U$ of dogfennel plants in the understory of this study site being approximately 1.3 mm.
This $S_U$ estimate agrees with the reductions of $P'_{T,o}$ below dogfennels: for example, mean $P_T$:$P'_{T,o}$ was 76.6% for rain-
only storms (Table 2), or a mean yield of $P_T = 12.9$ mm which exceeds a 1.3 mm reduction (due to $S_U$ and evaporation)
in the estimated mean $P'_{T,o}$ yield, 16.5 mm (from Table 1). A large portion of the rainwater captured on dogfennel
canopies was able to overcome stem water storage capacity and generate $P_S$. Dogfennel $P_S$ data were highly skewed,
producing a mean relative $P_S$ ($P_S$:$P'_{T,o}$) of 36.8%, but a median of 7.6% within a narrow interquartile range, 2.8%-
27.2% (Table 2). For events including occult precipitation, both maximum $P_S$:$P'_{T,o}$ and $P_T$:$P'_{T,o}$ exceeded 100%:
$P_T$:$P'_{T,o}$ during mixed storms maximized at 192%; whereas, the maximum for $P_S$:$P'_{T,o}$ was just over 900% (Table 2).
Note that dew in the understory was not measured by the above-canopy rainfall gauges and $P'_{T,o}$ was only increased
by an assumed maximum dew contribution equal to $S_U$ (1.33 mm), thus dew accumulation allows $P_T$ and $P_S$ to exceed
100% of $P_g$ and $P'_{T,o}$ (Table 2). When compared to rainfall above the overstory ($P_g$), the medians are much smaller:
$P_T$:$P_g$ being 45% and 58% for rain-only storms and mixed storms, respectively, and $P_S$:$P_g$ being 4.1% and 14.7%,
respectively (Table 2).
Yield [mm] were estimated for dogfennel $P_T$ and $P_S$ across storms, and both event-level $P_T$ and $P_S$ yields
linearly correlated with estimated event-level $P'_{T,o}$ (Figure 5a-b). Since, for $P_T$, the catchment area (canopy area above
the gauge) is equal to the input area (soil area below the gauge), $P_T$ yield from the canopy and $P_T$ supply to the surface
are equal and the term "yield" will be applied for both. Median $P_T$ yield beneath dogfennel for the measured storms
was 4.4 mm with an interquartile range of 1.1 mm to 11.3 mm (Figure 5c). Maximum $P_T$ yield approached 50 mm
during a large-magnitude rain storm (where $P_g = 101.3$ mm). Since the canopy area that generates stemflow is many
times greater than the surface area around plant stems that receive stemflow (see Table 1), $P_S$ yield and $F$ will differ.
$F$ are typically used to represent $P_S$ supply to soils, and is done so in the proceeding section. Yields of $P_S$ from
dogfennel were as high as 24 mm, but the median was 0.4 mm and the interquartile range was narrow, 0.1-1.3 mm
(Figure 5c).

**3.3. Stemflow and throughfall variability**

Coefficients of variability (CV) and quartile variability (CQV) were computed for both $P_S$ and $P_T$, relative to $P_g$ and $P'_{T,o}$ (Table 2), and storm-normalized temporal stability plots were generated for $P_S$ yield only (Figure 6). Storm-normalized temporal stability plots were not generated for $P_T$ yields because the experimental design accounts for its spatial variability through deployment of large gauge areas (compared to dogfennel canopy area); which permit estimates of variability across a few large-area gauges (Table 2), but limits the observable variability. CV and CQV for relative $P_T$ ranged from 22-90% and were generally lower for rain-only storms, <40%, than for mixed storms, >60% (Table 2). Variability in relative $P_S$ across study plants, ranging from 77-257%, was always greater than observed for relative $P_T$ for the monitored storms (Table 2). Due to the greater skew in the relative $P_S$ data compared to relative $P_T$, CV was many times greater than CQV for relative $P_S$ (Table 2). CV and CQV for $P_S:P'_{T,o}$ was similar for rain and the mixed storms; however, the CV for $P_S:P_g$ was greater for rain-only storms compared to mixed storms.

Temporal stability of normalized stemflow, $\bar{P}_{S,i}$ (Figure 6) indicates that there were only a few plants that captured most of the $P_{T,o}$ drained as stemflow (three plants' mean $\bar{P}_{S,i} \gg 1$). Thus, most of the studied dogfennel plants captured similar amounts of $P'_{T,o}$ as stemflow—having $\bar{P}_{S,i}$ between -1 and 1 (y = 0 represents the central tendency of $\bar{P}_{S,i}$ data). Funneling ratios ($F$ based on $P'_{T,o}$) show that all plants concentrated $P_S$ yields to the surface around their stem bases (Figure 6). Mean $F$ across all plants was 87, and for the 27 plants whose mean $\bar{P}_{S,i}$ fell between -1 and 1, median $F$ ranged 18-200 (Figure 6). However, for the three plants with the highest $\bar{P}_{S,i}$, their mean $F$ values were 287, 476 and 484 (Figure 6). These voluminous stemflow-generating plants, alone, account for one-third of total $P_S$ volume (8,734 mL / 27,870 mL). To evaluate possible canopy structural influences over $P_S$ variability, various directly-measured structural metrics were compared: radii of canopies and stems and the vertical variability in leaf angle (see supplemental Figure S4). No clear visible or statistical correlations or correspondences were found between these structural variables and $\bar{P}_{S,i}$ across plants (Figure S4). In fact, variability in the measured canopy structural variables was low (Table 1) compared to the variability observed for dogfennel $P_S$ and $\bar{P}_{S,i}$ (Figure 6).

**4. Discussion**

**4.1. Overstory throughfall partitioning by dogfennel**

Partitioning of overstory throughfall by this example dominant understory and pasture forb resulted in hydrologically relevant losses of rainwater to the surface at our site (Table 2). As maximum water storage capacity is a major driver of rainfall interception (Klaassen et al., 1998), the magnitude of dogfennel's overstory throughfall interception may be attributed to its canopy being able to store a sizeable magnitude of rainwater per unit area, 1.33 mm (Figure 4). Although mass changes of dried-and-submerged vegetation samples are discrepant from the processes and temporal scales of natural rainfall interception, it is a common method with well-known and long-discussed limitations selected to estimate water storage capacity since more direct water storage capacity estimation methods are still under development to date—see discussions in reviews by Friesen et al. (2015) and Klamerus-Iwan et al. (2020). Methodological limitations withstanding, the $S_U$ estimates in this study fit within the range of water storage capacities

of other herbaceous plants synthesized by Breuer et al. (2003). This synthesis is focuses on the leaves of herbaceous plants (alongside other plant types) (Breuer et al., 2003), but less research has estimated the stem component (or a reported a total including the stem component) of water storage capacity for short vegetation (Bradley et al., 2003; Wang et al., 2016; Wohlfahrt et al., 2006; Yu et al., 2012). The stems of herbaceous plants, even thick smooth stems (>1 cm in diameter) can store nearly 0.5 mm: e.g., *Taraxacum officinale* (dandelion) (Wohlfahrt et al., 2006). Even thin (<1 cm radius) herbaceous stems with epidermal outgrowths, like hairs, can store large amounts of rainwater: e.g., 0.25 mm for *Achillea millefolium* (yarrow) and 0.20 mm for *Trifolium pretense* (red clover) (Wohlfahrt et al., 2006). In the case of dogfennel stem water storage capacity at our site, the 0.43 mm estimate is within this range and its magnitude is likely a result of two principal factors: (1) dense stem coverage by desiccated leaves (photo in Figure 4); and (2) this species can achieve large densities, up to 700,000 stems ha$^{-1}$ (Dias et al., 2018) – 56,770 stems ha$^{-1}$ at our study site. We note that, to our knowledge, stem water storage capacities for herbaceous plants with spines, thorns, etc. have not been evaluated.

Overstory throughfall was also redistributed into a highly spatially variable (Table 2), but temporally persistent pattern beneath dogfennel canopies (where CV or CQV was approximately 20-40% for $P_T$ and 80-250% for $P_S$: Table 2), despite all measured canopy structures—like branch angle, stem size, canopy size, etc—being similar (Table 1). Since our sampling plan measured $P_T$ over a large area of the dogfennel canopy (rather than at numerous localized points), this discussion point will focus on the intraspecific $P_S$ observations. The high spatial variability and temporal persistence of $P_S$ across plants despite canopy structural similarity, raises the question: What caused the intraspecific $P_S$ patterns observed in this study? A likely explanation may be that, in this case, access to precipitation for stemflow production is related to overstory throughfall patterns (which, we reiterate, were not able to measure without removing or disrupting $P_T$ and $P_S$). Overstory throughfall patterns are well-known to be spatially variable, but temporally persistent across forest types (Van Stan et al., 2020). Specifically, individual dogfennel plants that persistently generated greater $P_S$ than other plants may have just received greater overstory throughfall from persistent overstory drip points. If the overstory throughfall pattern is a major driver of intraspecific variability in $P_S$ in this study, then the funneling ratios computed from mean overstory throughfall (per Figure 3) would be incorrect (in Figure 6). In this case, funneling ratios (computed from the localized overstory throughfall above each plant) could be similar across the monitored dogfennels. Testing this hypothesized relationship between dogfennel $P_S$ patterns and overstory throughfall patterns was not possible in the field, since sampling overstory throughfall would prevent $P_S$ from being generated by the plant. Future work to test this hypothesis could, however, make use of rainfall simulators.

The large diversion of rainwater and dew to their stem base may be partially responsible for dogfennel survival during extended periods of drought (or improved invasion efficacy during droughts: Loveless, 1959; Forthman, 1973), and may also explain why this species tends to be one of the most problematic in improved grazing systems located in Florida (Sellers et al., 2009). Rainfall patterns in central and south Florida may also intersect with dogfennel's canopy water balance to "tip the scales" in its favor. Specifically, rainfall in our study region is often limited from January through May, with the bulk of rainfall occurring from June through October, and the water storage capacity of burgeoning dogfennel plants during early spring may enhance chances of individual plant survival (resulting in large infestations as referenced previously).

**4.2. Overstory (woody) and understory (herbaceous) canopies may partition rainfall differently**

The dominant understory plant at our study site, dogfennel, intercepted similar amounts of modelled overstory throughfall, interquartile range 11-59% storm$^{-1}$ (Table 2), as compared to the gross rainfall interception by their overstory pine canopy, interquartile range 19-60% storm$^{-1}$ (Van Stan et al., 2017b). Similar rainwater interception between dogfennel and the pine overstory may be due to dogfennel's maximum water storage capacity comparing favorably to that of overstory tree species, 0.07-4.30 mm (Klamerus-Iwan et al., 2020). Even the maximum stem water storage capacity is of similar magnitude to values reported by past work on woody plants, 0.2-5.9 mm (Klamerus-Iwan et al., 2020), albeit on the lower end of the range. Most current research on stem water storage has focused on intrinsic factors of woody plant stems, like bark thickness, porosity, microrelief, or roughness (Ilek et al., 2017; Levia and Herwitz, 2005; Levia and Wubbena, 2006; Sioma et al., 2018; Van Stan et al., 2016; Van Stan and Levia, 2010); however, other stem structures besides bark may be capable of storing substantial water: e.g., the desiccated leaves of our study plant.

There were differences in how gross rainfall was redistributed by the overstory canopy compared to how modelled overstory throughfall was redistributed by the dogfennel understory. Stemflow from the overstory, *P. palustris*, was negligible at this site, 0.2% of gross rainfall (Yankine et al., 2017), but median dogfennel $P_S$ was 7.6% of modelled overstory throughfall (with an interquartile range of 2.8-27.2%) (Table 2). Annual relative $P_S$ (and $P_T$) estimates from trees and herbaceous plants reported by previous work indicates that herbaceous plants are generally greater stemflow producers than woody plants (Sadeghi et al., 2020). Although relative $P_T$ beneath dogfennel was similar to observations of relative overstory throughfall beneath *P. palustris* at this site (Mesta et al., 2017), throughfall has been found to be generally lower beneath herbaceous plant canopies than for woody ones (Sadeghi et al., 2020). This seems reasonable, because, if interception is similar between herbaceous plants and woody plants, then an increase in relative stemflow would necessitate a decrease in relative throughfall. The results of this study support statements by several past studies suggesting that plants in the understory and overstory interact differently with rainfall. Thus, we repeat the long-standing calls for greater research on understory precipitation partitioning, particularly stemflow, research (Price et al., 1997; Price and Watters, 1989; Verry and Timmons, 1977; Yarie, 1980).

**4.3. A brief discussion on dew-generated throughfall and stemflow**

For a few storms ($n = 5$), dew contributed significantly to $P_T$ and $P_S$ by the studied dogfennel plants. The median $P_T$ generated from dew beneath dogfennels at our site was 0.74 mm plant$^{-1}$ with an interquartile range of 0.47-0.99 mm plant$^{-1}$, resulting in a total dew-related contribution to T of 17.1 mm over the study period. Volumes of stemflow under dewfall totaled 558 mL for all study plants, with individuals supplementing the dew-related $P_T$ with up to 61 mL plant$^{-1}$ (yielding an additional ~0.6 mm). Dew contributions to net precipitation below plant canopies have rarely been studied. The earliest quantity for dew drainage was 0.08 mm from a single event on a single tree in Johanniskreuz, Germany (Ney, 1893). Since then, to our knowledge, only one other study has examined dew-related drainage from plants, focusing on stemflow from the herbaceous *Ambrosia artemisiifolia* (common ragweed) (Shure and Lewis, 1973). They estimated that the drainage of dew via $P_S$ resulted in an additional input of 1.1 L month$^{-1}$ during the growing season, and hypothesized that this process may "play a vital role in governing the density, diversity, and

distribution of plant species within field ecosystems" (Shure and Lewis, 1973). Dew drainage from plant canopies and
down stems may, in addition to being a valuable water source, influence plant-soil interactions by transporting leached
or dry deposited materials to the soils—something also discussed by Shure and Lewis (1973). Globally, dew
contributes a small percentage to the annual precipitation (Baier, 1966), however, in semiarid and arid (Baier, 1966;
Hao et al., 2012), as well as summer-dry climates (Tuller and Chilton, 1973), dew can form a significant water input.
It is reasonable to suppose, then, that in such ecologic settings as these any factor which doubles the frequency of
plant-moisture availability, even though the amounts be small, must materially affect the plant growing condition.
Therefore, further research is needed to assess dew (and mixed storms) drainage in arid and semiarid climates, with
days on which dew occurs being ≥70% per year (Hao et al., 2012). The global importance of occult precipitation and
resulting wet canopy conditions has recently been reviewed and described as a critical future research direction for
plant sciences (Dawson and Goldsmith, 2018). Given these scant but ecologically relevant findings, further research
on the influence of condensation events on plant-soil interactions via throughfall and stemflow may be merited.
**5. Conclusions**
*Eupatorium capillifolium* (Lam., dogfennel) in the understory of an urban forest fragment intercepted 20.4% of
modelled overstory throughfall from *Pinus palustris* (Mill.). The remaining 71.0% and 7.9% of modelled overstory
throughfall reached the surface beneath dogfennels as understory throughfall and stemflow, respectively. At the stand
scale, the partitioning of modelled overstory throughfall by this understory forb differs considerably from the rainfall
partitioning of the woody overstory, especially regarding stemflow (7.9% versus <0.2%). During a few storms that
occurred in tandem with dewfall, dogfennels were able to augment stemflow (and throughfall) production through
capturing dew. These processes may help explain how dogfennels survive extended droughts, and even show improved
invasion efficacy during droughts, making it one of the most problematic weeds in southeastern US grazing systems.
Stemflow variability among individual plants was very high (CV ~250%), but no dogfennel canopy structures
measured in this study provided statistically significant insights into this stemflow variability. Future work will assess
to what extent actual overstory throughfall variability drives understory stemflow variability for plants, like dogfennel,
of similar intraspecific canopy structure. The inability to measure fine-scale overstory throughfall patterns without
disturbing understory rainfall partitioning in the field is a non-trivial limitation of this study—a limitation that future
work may overcome with rainfall simulations. Still, in forests, overstory throughfall is not the final frontier for
determining net rainfall, and investigations on how it is intercepted and redistributed by herbaceous plants is needed
to improve our understanding of exactly how much (and in what pattern) rainfall reaches the surface. For other
vegetated ecosystems where herbaceous plants are the overstory (grasslands and croplands), precipitation partitioning
research is also needed.
**Acknowledgements**
DARG acknowledges support from the US Dept of Education Ronald E. McNair Program and AMJC acknowledges
support from NWO Earth and Life Sciences (ALW), veni-project 863.12.022.

**Code/Data availability**

Data is permanently archived at https://digitalcommons.georgiasouthern.edu/ and freely available.

**Author contribution**

DARG conceived and designed the study in consultation with JTVS and AMJCG. DARG designed field collection devices in consultation with JTVS and AMJCG, then deployed devices, collected data, performed the data analysis, and drafted the initial manuscript with input from all authors. BAS contributed expertise regarding relevant range- and pastureland topics, assisting with data analysis/interpretation. SMMS performed a literature synthesis for discussions comparing herbaceous and woody plants' rainfall partitioning and used this synthesis to assist in manuscript writing. JTVS was the principal undergraduate research supervisor for DARG. All authors contributed to manuscript revisions.

**Competing interests**

The authors have no competing interests.

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

**Table 1:** Descriptive event statistics for rainfall (observed), overstory throughfall (estimated per Figure N) and
measured individual plant traits. When minimum overstory throughfall was zero, dew occurred – as verified by air
temperatures equalling dew point temperatures.

| Parameter (units) | Mean | Median | ±SD | Min. | Max. |
|---|---|---|---|---|---|
| Rainfall (mm) | 16.5 | 6.6 | 25.8 | 0.1 | 101.3 |
| Overstory throughfall (mm) | 11.0 | 3.5 | 18.7 | 0.0 | 72.2 |
| Canopy radius (cm) | 18.3 | 18.4 | 4.5 | 12.2 | 26.2 |
| Stem radius (cm) | 0.5 | 0.6 | 0.1 | 0.3 | 0.7 |
| Canopy:stem radii | 36.3 | 36.1 | 7.4 | 24.1 | 50.0 |
| Leaf angle at the stem (degrees from vertical) | | | | | |
|    1.00 m height | 54.0 | 54.0 | 2.0 | 50.5 | 59.0 |
|    1.25 m height | 45.9 | 46.5 | 3.1 | 40.5 | 50.5 |
|    1.50 m height | 39.6 | 39.5 | 1.8 | 36.0 | 43.0 |
|    1.75 m height | 34.0 | 34.5 | 2.3 | 30.0 | 39.0 |
|    2.00 m height | 31.9 | 32.0 | 2.8 | 25.0 | 36.5 |


**Table 2:** Descriptive statistics of relative throughfall ($P_T$) and stemflow ($P_S$) yield from dogfennel plants expressed as a proportion of gross rainfall ($P_g$) and estimated overstory throughfall ($P_{T,o}$). Coefficients of variation (CV) and quartile variation (CQV) are also provided. For storms where dew occurred in the understory, dew was not measured by above-canopy $P_g$ gauges, but was included in the estimated $P_{T,o}$ estimate by assuming dew represented at least additional 1.33 mm (i.e., $S_u$).

| Parameter | Mean (SD) | Median | Q1 | Q3 | Max | CV | CQV |
|---|---|---|---|---|---|---|---|
| Rain storms | | | | | | | |
| $P_T$:$P_g$ (%) | 43.6 (15.2) | 44.9 | 34.3 | 52.4 | 101.7 | 34.9 | 20.9 |
| $P_S$:$P_g$ (%) | 18.8 (47.3) | 4.1 | 1.7 | 13.8 | 434.3 | 251.6 | 78.1 |
| $P_T$:$P_{T,o}$ (%) | 76.6 (29.3) | 72.0 | 58.5 | 91.1 | 190.6 | 38.3 | 21.8 |
| $P_S$:$P_{T,o}$ (%) | 36.8 (93.5) | 7.6 | 2.8 | 27.2 | 900.3 | 254.1 | 81.3 |
| Mixed storms* | | | | | | | |
| $P_T$:$P_g$ (%) | 70.3 (43.7) | 58.0 | 39.5 | 102.9 | 149.4 | 62.2 | 44.5 |
| $P_S$:$P_g$ (%) | 32.7 (45.2) | 14.7 | 5.2 | 39.7 | 198.0 | 138.2 | 76.8 |
| $P_T$:$P_{T,o}$ (%) | 72.0 (30.2) | 69.1 | 53.2 | 86.9 | 191.6 | 41.9 | 24.1 |
| $P_S$:$P_{T,o}$ (%) | 33.4 (86.2) | 8.1 | 3.0 | 24.3 | 900.3 | 257.4 | 78.0 |

*Storms with occult precipitation.

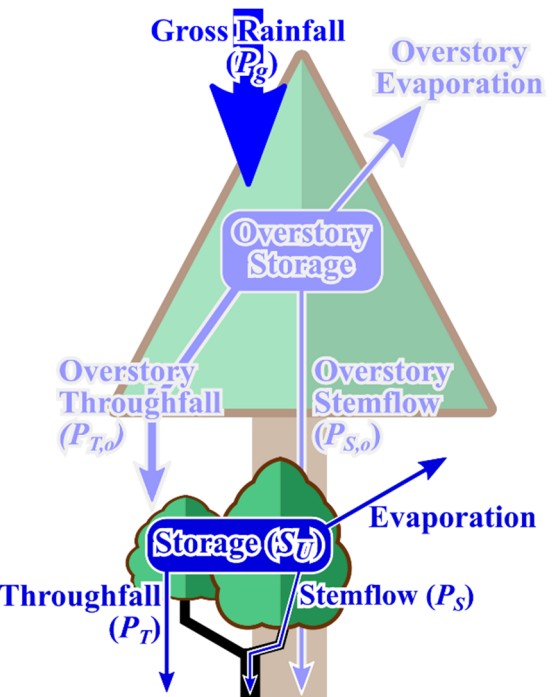


**Figure 1:** Partitioning of gross rainfall by the overstory (light blue) and by the understory (dark blue). Overstory
throughfall ($P_{T,o}$), the input to the understory canopy, was estimated from past work at the site (see supplemental
materials). In this study, overstory throughfall was modelled ($P'_{T,o}$ per Methods Section 2.2.2.) and maximum
understory water storage capacity ($S_U$), throughfall ($P_T$), and stemflow ($P_S$) were measured.

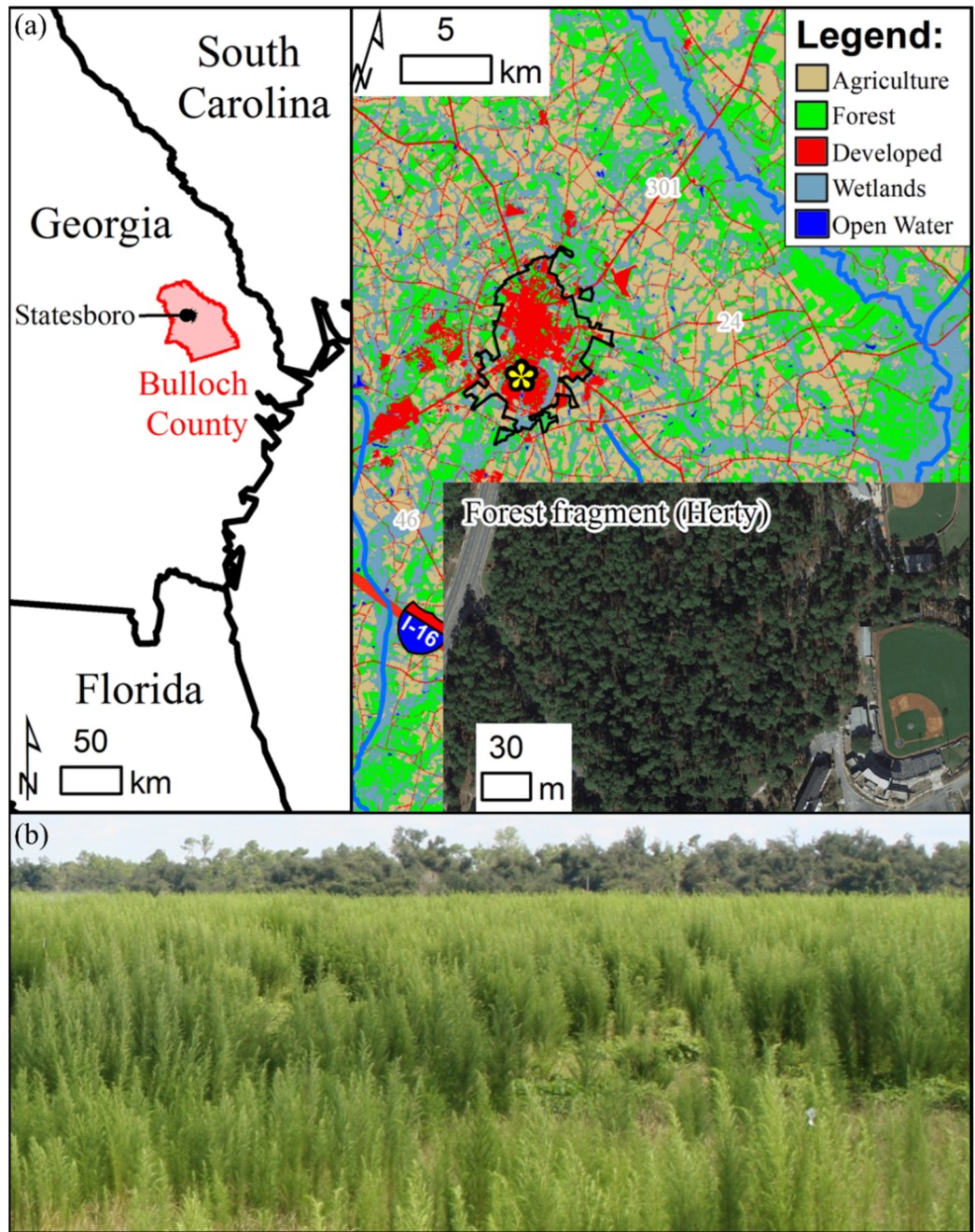

**Figure 2:** (a) Location of the studied *Pinus palustris* (longleaf pine) forest fragment, Charles H. Herty Pines Nature Preserve, on the Statesboro, Georgia (USA) campus of Georgia Southern University, where *Eupatorium capillifolium* (dogfennel) is a dominant understory plant. (b) Dogfennel can dominate pastures as well, as shown by the photograph (credit: Brent A. Sellers). Map layer sources: State and county boundaries, and aerial imagery ©ESRI, TomTom North America, Inc. The land use layer was derived from the National Land Cover Database 2011 (full metadata and data access link: https://gdg.sc.egov.usda.gov/Catalog/ProductDescription/NLCD.html).

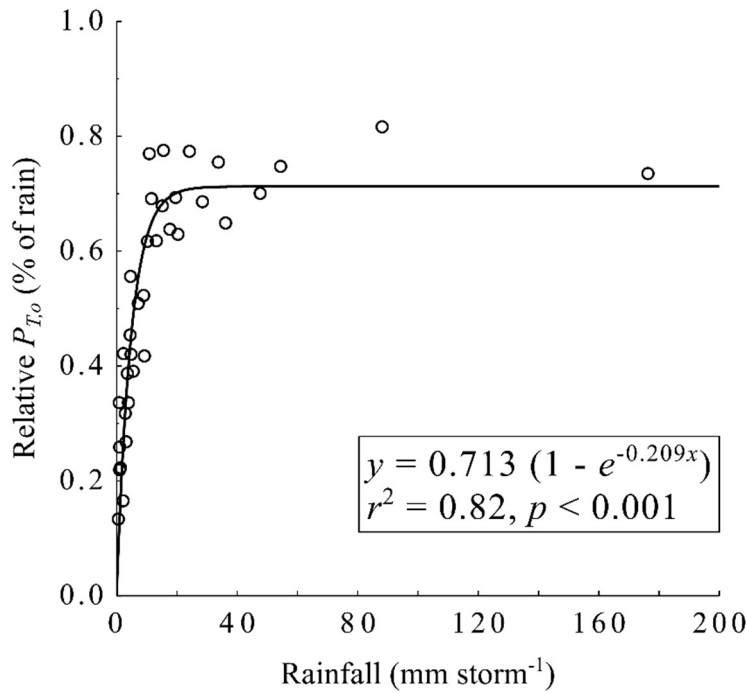


**Figure 3:** Observed relative overstory throughfall ($P_{T,o}$) in relation to above-canopy rainfall at the study site.

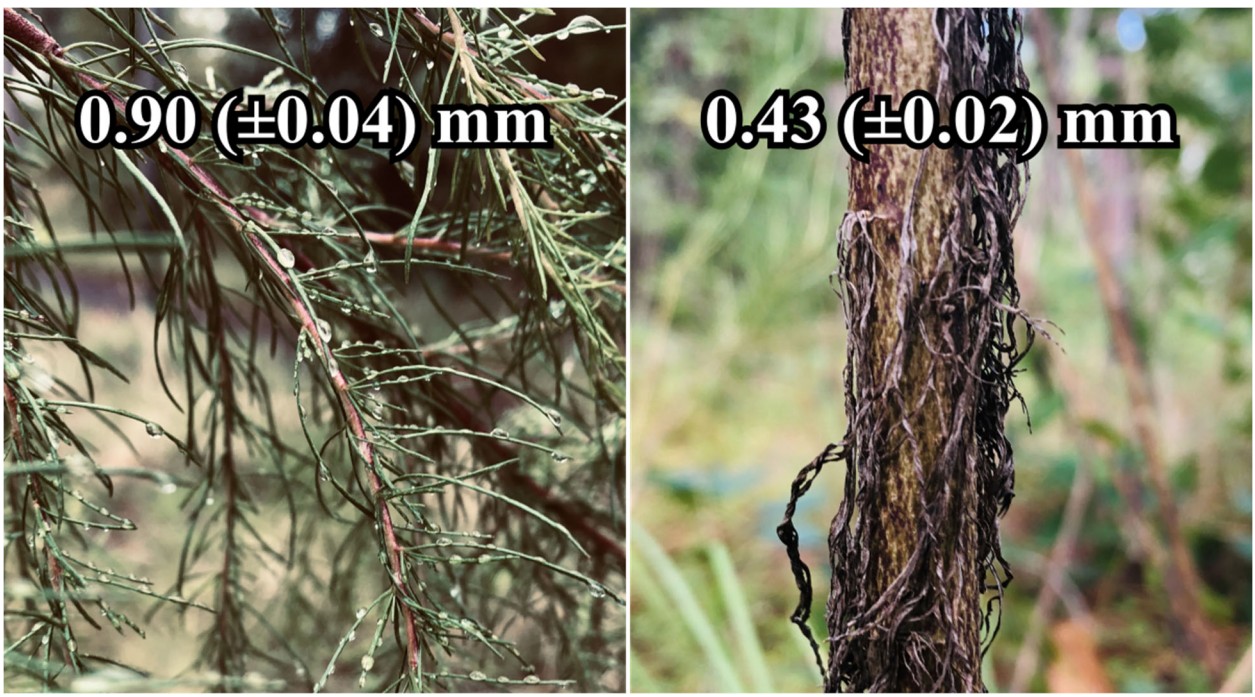

**Figure 4:** Water storage capacity (standard error) for the (left) canopy and (right) stem of *Eupatorium capillifolium* (dogfennel) per lab-based submersion tests on samples collected from the Herty Pines understory.

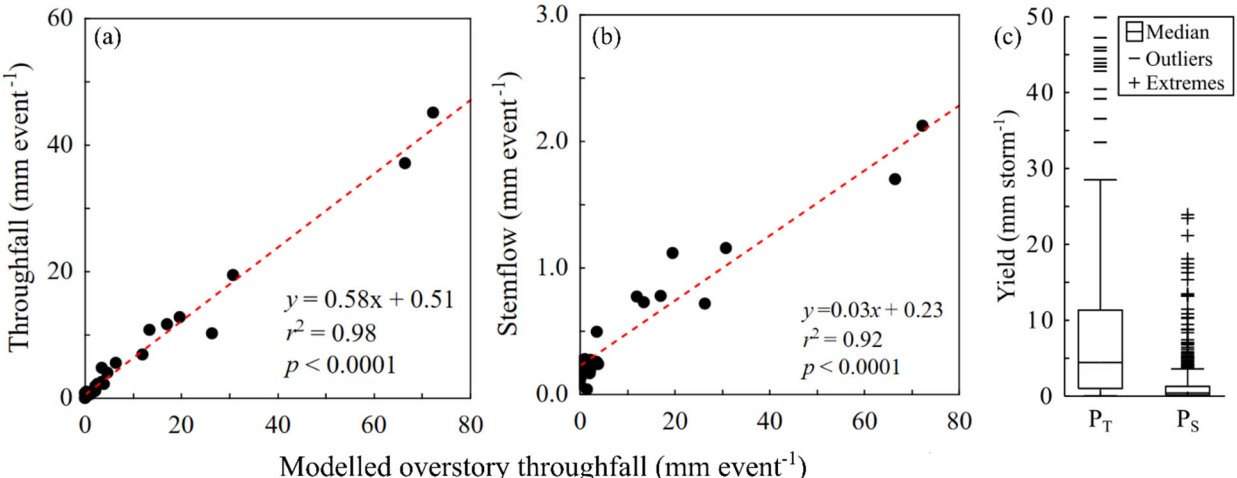


**Figure 5:** Scatter plots showing the response of *Eupatorium capillifolium* (dogfennel) (a) throughfall ($P_T$) and (b)
stemflow ($P_S$) yields across all rainfall events (without occult precipitation). (c) Boxplot showing yields from
individual $P_T$ gauges and plants' $P_S$ (Line and box: median and interquartile range; whiskers: non-outlier range; other
symbols represent outliers and extreme values).

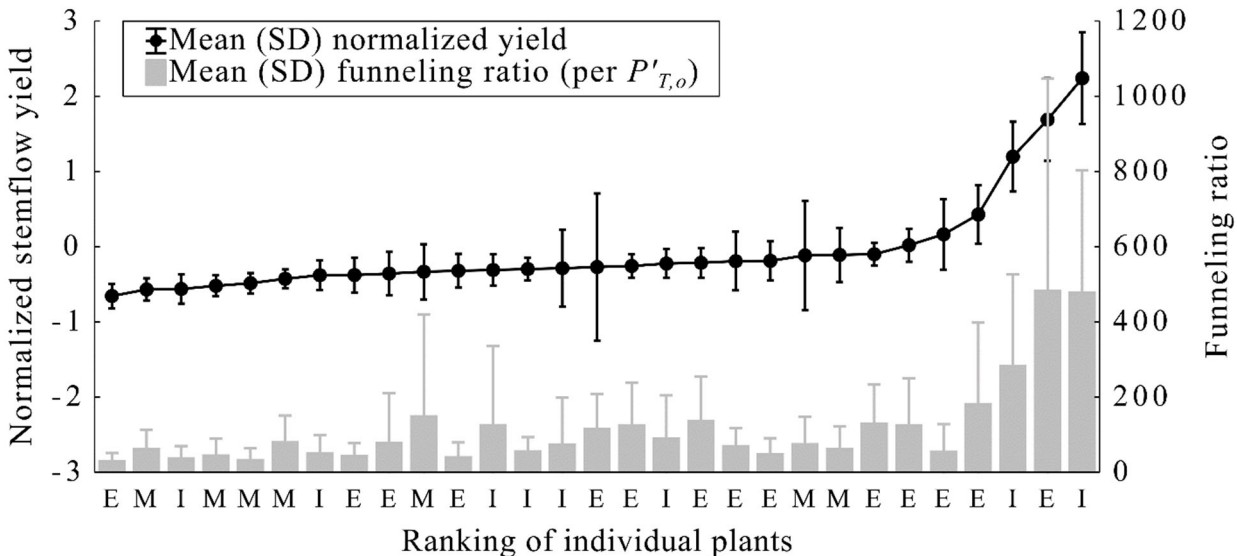


**Figure 6:** Mean and standard deviation (SD) of normalized stemflow yield plant[-1] and associated funneling ratio per
Herwitz (1986) and using modelled overstory throughfall ($P'_{T,o}$) in order of rank per mean normalized stemflow yield.
Plant locations within clusters are indicated (E = external, M = middle, between the interior and exterior, and I =
interior).