# Peer review of "Rainfall interception and redistribution by a common North"

_Hydrology and Earth System Sciences, 2019_

## Referee Comment (RC1) · Anonymous Referee #1 · 11 Nov 2019

Gordon et al. present observations of understory rainfall interception in the southeastern US. Dogfennel, the understory plant they study, is a tall and dense forb that the authors show can have a major effect on rainfall partitioning. The topic is of great interest to HESS readers, as rainfall interception is an important component of the water cycle that is nevertheless relatively poorly studied and represented in models. The authors make a compelling case that this is particularly so for low-stature and understory vegetation, such as the dogfennel communities they study. The manuscript, however, does not quite live up to the expectations raised in the powerfully argued introduction.

[Figure]

The analyses and interpretations contain considerable flaws and omissions: a flux that is essential to the authors' conclusions was interpolated rather than directly measured, which is only addressed in a cursory manner; the overall partitioning is never (or if so, inappropriately) estimated; and important details regarding the methods and observations are missing.

Event-level overstory throughfall

The overstory throughfall fluxes, which act as the normalization factor in the most important event-level rainfall partitioning estimates and are thus essential to the authors' conclusions, are interpolated rather than measured. The authors acknowledge this potentially major source of uncertainty only briefly when discussing the spatial variability (l280, Fig. 5). I feel that this issue needs to be addressed head on, as I have several concerns. First, it further introduces spatial variability. However, the spatial variability of understory throughfall (and overstory throughfall) is not analysed, despite the redundancy in the measurements. Second, there could well be an association between overstory throughfall and relevant dogfennel parameters such as their density. Such an association would need to be addressed if the authors want to draw robust ecosystem-level conclusions. Third, there may be a temporal bias here as well, but it is difficult to say because the study periods in which overstory and understory throughfall measurements were conducted are neither stated nor compared. Fourth, while the interpolation of spatially averaged overstory throughfall (supplement) provides a decent fit overall, the linear association is clearly insufficient for small rain events. For zero rainfall, it predicts negative throughfall. The authors, however, analyse small events in great detail. It is not clear to me how these issues impinge on their estimates of understory throughfall for small events. Similarly, the uncertainties that arise from the ad-hoc estimation of dewfall are not addressed or quantified. In summary, I thus have major concerns regarding the estimated event-level understory water balance, especially during small events.

Insufficient estimates of total rainfall partitioning

Apart from the issues raised above, the statistical analyses of rainfall partitioning are insufficient. The authors do not report the overall partitioning over the entire study period (e.g. stemflow vs total rainfall or overstory throughfall) and the associated uncertainties. The analyses at the event level that are shown are insufficient for three reasons. First, the overall partitioning is not reported. The individual ratios (e.g. stemflow divided by rainfall) in Tab. 1 cannot be averaged to obtain the overall ratio. The authors, however, do just that in the conclusions (they even report the median rather than the average) when they write: 'Eupatorium capillifolium (Lam., dogfennel) in the understory of an urban forest fragment intercepted 20.4% of verstory throughfall from Pinus palustris (Mill.).. I would expect both errors (aggregating ratios, median instead of mean) to overemphasize small events, and thus to overestimate throughfall/rainfall. The event-level fluxes need to be summed, see e.g. doi:10.1029/2000WR900074, doi:10.1016/S0022-1694(01)00393-6, doi:10.1088/1748-9326/ab1049, for how to estimate overall partitioning and its uncertainties (due to spatial variability, stems that were not instrumented, observation errors, etc.). Second, only summary statistics such as the median are shown (Tab. 1). A scatter plot would allow the reader to draw additional inferences, such as in what way stemflow increases disproportionately for larger events. Third, it is not clear how the data were spatially aggregated. Three clumps were instrumented, and I assume they were averaged over, but how?

Interception capacity

I could not follow the rationale behind the interception capacity measurements. How long were they dried in the oven? Did the leaf itself (not the intercepted water) lose weight during that period? Why not compare it to the weight before wetting? The other issue, which is that the submersion in the lab is very different from the wetting due to rainfall in the field, would remain. This needs to be spelled out clearly, cf. doi:10.1016/S0022-1694(01)00393-6.

Missing details

[Figure]

Several crucial aspects of the observations and analyses are not addressed in the methods:

The throughfall funnels are not described in detail, and there is not a single picture. In particular, I could not find relevant information on whether they were adequate for measuring below-dogfennel throughfall. The authors argue that they provide more robust estimates because they are larger than most rain gauges that are commonly used for such purposes, but at approximately 25x25 cm, this difference does not strike me as particularly noteworthy. Given the relatively large density of dogfennel plants, however, it is not clear to what extent the plants and hence the throughfall were disturbed by the installation of the funnels.

I would not be able to reproduce the scaling of the rainfall interception capacity measurements from the leaf to the plot scale. The authors mention in l176 that they use estimates of leaf area, but these estimates are never introduced. Equations would also help, as would a consistent terminology (surface area seemingly refers to very different things in the same paragraph).

It is not clear how dogfennel density (e.g. at what scale) was determined and whether the numbers given in Section 2.1 refer to the clumps the authors study or to other areas.

The three clumps the authors study are not described in detail. How do they differ? What do they look like? Does that have an impact on the rainfall partitioning?

The regression analysis shown in Fig. 5 (of doubtful value because it relies on the unrealistic assumption on overstory throughfall) is not described. According to the figures, it looks like the authors regressed the ranks rather than the actual observations, which would need justification. So would the fact that the authors apparently did not consider the joint influences of explanatory variables.

Minor points:

Figure 6 compares rainfall partitioning of herbaceous plants and trees, but I suppose the climatic conditions differ between the two and thus constitute a major confounding factor. These concerns are, however, not addressed.
* * *

---

## Referee Comment (RC2) · Anonymous Referee #2 · 13 Mar 2020

Manuscript "Rainfall interception and redistribution by a common North American understory and pasture forb, Eupatorium capillifolium(Lam.  dogfennel)" by D. Alex R. Gordon, Miriam Coenders-Gerrits, Brent A. Sellers, S.M. Moein Sadeghi, John T. and Van Stan II.

This manuscript investigates interception and redistribution of precipitation related to herbaceous plants, i.e.  understory forbs growing in an urban disturbed pine forest, resp. in pastures. Partitioning of overstory throughfall into understory throughfall and stemflow is analysed as well as water storage capacities for leaves and stems. Among

others, it was found that the investigated dogfennel plants can capture and drain dew to the basis of their stems, being effective for enduring drought periods.

In my view, the submitted study is highly interesting and convincing, carefully conducted, and well presented. It is well suited for the journal and should be of high interest for the readers.

For further clarification, I suggest to add more details on the conducted lab investigations: how long were the plant samples dried in the oven, and more details on the mass balances (how the final values of storage capacity were obtained) would be helpful.

—————————————————————

---

## Author Comment (AC1) · 7 Apr 2020

1) "Gordon et al. present observations of understory rainfall interception in the southeastern US. Dogfennel, the understory plant they study, is a tall and dense forb that the authors show can have a major effect on rainfall partitioning. The topic is of great interest to HESS readers, as rainfall interception is an important component of the water cycle that is nevertheless relatively poorly studied and represented in models. The authors make a compelling case that this is particularly so for low-stature and understory vegetation, such as the dogfennel communities they study."

Response 1) We thank Reviewer 1 for their appreciation of the manuscripts' strengths and insightful comments regarding the study weaknesses. We have addressed these comments as described below and believe our revised manuscript has been greatly improved.

2) "The overstory throughfall fluxes, which act as the normalization factor in the most important event-level rainfall partitioning estimates and are thus essential to the authors' conclusions, are interpolated rather than measured. The authors acknowledge this potentially major source of uncertainty only briefly when discussing the spatial variability (l280, Fig. 5). I feel that this issue needs to be addressed head on, as I have several concerns. First, it further introduces spatial variability. However, the spatial variability of understory throughfall (and overstory throughfall) is not analysed, despite the redundancy in the measurements. Second, there could well be an association between overstory throughfall and relevant dogfennel parameters such as their density. Such an association would need to be addressed if the authors want to draw robust ecosystem level conclusions. Third, there may be a temporal bias here as well, but it is difficult to say because the study periods in which overstory and understory throughfall measurements were conducted are neither stated nor compared. Fourth, while the interpolation of spatially averaged overstory throughfall (supplement) provides a decent fit overall, the linear association is clearly insufficient for small rain events. For zero rainfall, it predicts negative throughfall. The authors, however, analyse small events in great detail. It is not clear to me how these issues impinge on their estimates of understory throughfall for small events."

Response 2: The reviewer is correct that we estimated the stand-level overstory throughfall flux using recent data measured at the site. Unfortunately, it is not possible to directly measure overstory throughfall AND measure understory partitioning simultaneously. This is because the direct measurement (via tipping buckets or bottles, etc) of overstory throughfall would disturb or remove understory throughfall and stemflow. We respectfully disagree that we have "only addressed [this issue] in a cursory manner." In

fact, we explicitly state in lines 133-138 (and in the supplement) how and why we estimated overstory throughfall, and discussed how this constrains our ability to broadly interpret understory throughfall and stemflow patterns. The reviewer raises two issues regarding the overstory throughfall estimation methods: (i) the linear association between rainfall and overstory throughfall "predicts negative throughfall" in small events; and (ii) there "may be a temporal bias" as the data for estimating overstory throughfall were collected prior to the start of this study. Regarding point (i), the reviewer is absolutely correct. However, overstory throughfall estimates for our small storms were not negative; so, after returning to the analysis spreadsheet, we realized that the wrong method was reported. Overstory throughfall (TF_o) was estimated from the association between TF_o (as a % of rainfall) and storm size (R) using the so-called "Aston" curve: TF_o[%] = a * (1 - EXP (-b * R[mm])). This does not return negative TF_o for small storms. We apologize for this error and have updated the supplemental figure to reflect the correct method. Regarding point (ii), we do not believe that there is any significant temporal bias. The canopy is mature and there has been no known/noticeable disturbance or change in canopy structure. As a result, although one can never be entirely certain, we assume that the association between overstory throughfall and storm size has not changed. This is now explicitly stated in the methods in lines 135-138.

3) "the uncertainties that arise from the ad-hoc estimation of dewfall are not addressed or quantified."

Response 3: The dew estimation was, in fact, done post-hoc: after dew was observed during sampling. Still, we have edited the manuscript in lines 140-145 to explicitly state the conditions surrounding our post-hoc dew estimate, including: (1) the assumption underlying the dew estimates (equating it with canopy water storage capacity), (2) the implications of this assumption (that dew estimates are maximums), and (3) confirmation of dew occurrence using quantitative meteorological measurements (beyond the binary/qualitative present or absent, to the eye, during sampling).

4) "Apart from the issues raised above, the statistical analyses of rainfall partitioning

are insufficient. The authors do not report the overall partitioning over the entire study period (e.g. stemflow vs total rainfall or overstory throughfall) and the associated uncertainties. The analyses at the event level that are shown are insufficient for three reasons. First, the overall partitioning is not reported. The individual ratios (e.g. stemflow divided by rainfall) in Tab. 1 cannot be averaged to obtain the overall ratio. The authors, however, do just that in the conclusions (they even report the median rather than the average) when they write: 'Eupatorium capillifolium (Lam., dogfennel) in the understory of an urban forest fragment intercepted 20.4% of verstory throughfall from Pinus palustris (Mill.).. I would expect both errors (aggregating ratios, median instead of mean) to overemphasize small events, and thus to overestimate throughfall/rainfall. The event-level fluxes need to be summed, see e.g. doi:10.1029/2000WR900074, doi:10.1016/S0022-1694(01)00393-6, doi:10.1088/1748-9326/ab1049, for how to estimate overall partitioning and its uncertainties (due to spatial variability, stems that were not instrumented, observation errors, etc.). Second, only summary statistics such as the median are shown (Tab. 1). A scatter plot would allow the reader to draw additional inferences, such as in what way stemflow increases disproportionately for larger events. Third, it is not clear how the data were spatially aggregated. Three clumps were instrumented, and I assume they were averaged over, but how?

Response 4: We appreciate the reviewer's comments on the statistical analyses and, firstly, we agree that the overall median is not the standard statistic reported for annual precipitation partitioning fractions – it is the sum. Therefore, we now include the total precipitation partitioning fractions from scaled summations across the study. Secondly, the reviewer requested a scatter plot of event summed data with respect to storm size. This has been added to the manuscript (panels a-b in the revised Figure 4). Thirdly, regarding spatial considerations, no spatial analyses (beyond comparison of CV and normalized stemflow values) were done. Even these spatial analyses are rarely done (see recent review: doi: 10.1007/978-3-030-29702-2_6).

5) "I could not follow the rationale behind the interception capacity measurements.

How long were they dried in the oven? Did the leaf itself (not the intercepted water) lose weight during that period? Why not compare it to the weight before wetting? The other issue, which is that the submersion in the lab is very different from the wetting due to rainfall in the field, would remain. This needs to be spelled out clearly, cf. doi:10.1016/S0022-1694(01)00393-6.

Response 5: We agree with the reviewer that the water storage capacity estimation methods require clarification. The details requested by the reviewer (and some additional information) is now provided in Lines 178-188. We also agree that we should clarify differences between this method and how leaves/stems wet in nature. This has been added to the manuscript.

6) "The throughfall funnels are not described in detail, and there is not a single picture. The authors argue that they provide more robust estimates because they are larger than most rain gauges that are commonly used for such purposes, but at approximately 25x25 cm, this difference does not strike me as particularly noteworthy. Given the relatively large density of dogfennel plants, however, it is not clear to what extent the plants and hence the throughfall were disturbed by the installation of the funnels.

Response 6: Dimensions of funnels are described, and we now provided a photograph of a deployed throughfall gage in the supplemental materials. To clarify: we did not state that these funnels were bigger than most funnels – rather, we noted that funnel size was larger per unit canopy area for the studied plant, dogfennel, compared to trees.

7) "I would not be able to reproduce the scaling of the rainfall interception capacity measurements from the leaf to the plot scale. The authors mention in l176 that they use estimates of leaf area, but these estimates are never introduced. Equations would also help, as would a consistent terminology (surface area seemingly refers to very different things in the same paragraph)."

Response 7: We agree and have provided greater detail on the scaling methods for

water storage capacity in lines 195-202, stating "Specific water storage capacity estimates for the stem (0.436 mm) and leaves (0.195 mm) were then scaled to Su [mm as L m-2] using stem and leaf surface area estimates per plant (171.9 cm2 plant-1 and 807.5 cm2 plant-1, respectively), and multiplied by the site plant density (5.68 plants m-2) and divided by 1000. Plant stem and leaf surface area estimates were determined from 5 representative plants that were cut from the site and separated into leaves and stems, then the sum of leaf and stem areas (determined as mentioned earlier in the paragraph) were divided by 5."

8) "It is not clear how dogfennel density (e.g. at what scale) was determined and whether the numbers given in Section 2.1 refer to the clumps the authors study or to other areas."

Response 8: We agree. Details for estimating stem density are now provided in lines 115-117, stating "Dogfennel density was estimated in ten 10x10 m plots by counting the stems clump-1 for 3 randomly-selected clumps in each plot. For each plot, the mean stems clump-1 were multiplied by the number of clumps plot-1. Finally, all stems plot-1 were summed and scaled to 1 ha."

9) "The three clumps the authors study are not described in detail. How do they differ? What do they look like? Does that have an impact on the rainfall partitioning?"

Response 9: The plants from each clump are described in detail in Table 1. From the details in Table 1, the plants were all very similar.

10) "The regression analysis shown in Fig. 5 (of doubtful value because it relies on the unrealistic assumption on overstory throughfall) is not described. According to the figures, it looks like the authors regressed the ranks rather than the actual observations, which would need justification. So would the fact that the authors apparently did not consider the joint influences of explanatory variables."

Response 10: We have removed the bottom panels of Fig. 5 and now provide scat-
terplots in the supplement (new Figure S5) that shows no statistically significant correlations for the variables (or no significant differences for the categorical variable) presented in the old Figure 5. No multivariate statistical methods were applied to assess multivariate influences over stemflow variability as all bivariate results were very highly un-significant ($r2 \sim 0$ and $p > 0.9$).

11) "Figure 6 compares rainfall partitioning of herbaceous plants and trees, but I suppose the climatic conditions differ between the two and thus constitute a major confounding factor. These concerns are, however, not addressed."

Response 11: We have removed Figure 6 from the manuscript for two reasons (1) there is already a synthesis work published that we can cite and (2) we believe that this synthesis figure merits greater consideration in a different, broader paper.

---

## Author Comment (AC2) · 7 Apr 2020

1) "In my view, the submitted study is highly interesting and convincing, carefully conducted, and well presented. It is well suited for the journal and should be of high interest for the readers. For further clarification, I suggest to add more details on the conducted lab investigations: how long were the plant samples dried in the oven?, and more details on the mass balances (how the final values of storage capacity were obtained) would be helpful."

Response 1: We thank Reviewer 2 for their appreciation of the manuscript and their suggested revisions. We have addressed these comments by providing greater details on the lab investigations and related data analyses in lines 115-117, 135-138, 140-145, and 178-202; as well as by providing additional figures to the manuscript (Figure 4a-b) and the supplemental materials (Figures S2, S3b, and S5).

———————————————————

---

## Author Response (AR2)

**Responses to Reviewer 1**

"The presentation and clarity have improved. I want to highlight the description of the field measurements and the time-aggregate partitioning estimates, both of which are much more through now. However, two major limitations remain."

**Response: We again thank Reviewer 1 for their insightful comments regarding the remaining study weaknesses. In addressing these comments, we believe our manuscript has been improved.**

1) The water balance estimates are based on questionable and largely untested assumptions regarding overstory throughfall. It is neither clear nor explored how the neglected spatial variability of actual throughfall, as well as temporal changes and limitations in the curve fitting underlying the extrapolation, affect the conclusions. There are numerous potential systematic confounding factors at play. Furthermore, the overstory throughfall measurements that form the basis of the extrapolation are not well documented (AGU presentation), further limiting my ability to assess their validity. These issues continue to be downplayed throughout the manuscript. For instance, they are not addressed in the abstract.

**Response: We understand this concern and have revised the manuscript at various points to ensure the overstory throughfall estimates ($P_{T,o}$) are explicitly described:**

**(1)** **Overstory throughfall used in our study is now explicitly called "modelled overstory throughfall" and the parameter has been renamed to $P'_{T,o}$ so that the accent indicates that it has been modelled.**
**(2)** **A full description of how and when $P_{T,o}$ was measured and $P'_{T,o}$ was estimated is now provided in the Methods section, in its own dedicated subsection, where the supplemental figure is now placed, and its temporal variability is described.**
**(3)** **Also in the $P_{T,o}$ methods subsection we describe how the equation was derived.**
**(4)** **Limitations of using modelled overstory throughfall and its possible impacts on the results are further discussed: (a) in the methods alongside an explanation that it was not possible to simultaneously measure $P_{T,o}$, $P_T$, and $P_S$; (b) at the very start of the results; (c) in the discussion and conclusions; and (d) in the abstract so that, from the very start, the limitations and possible impacts of modelled overstory throughfall are made apparent to the reader.**

2) A further limitation in the data acquisition pertains to the estimates of the storage capacity. These are based on the mass change that chopped up samples that had been soaked in water for days underwent during oven drying. The processes and temporal scales involved are so discrepant from regular interception that they should be mentioned in the discussion.

**Response: We agree that our ex situ lab estimates of storage capacity (S) do not exactly match natural wetting processes, but no lab-based S estimation method developed to-date has provided an exact match to in situ wetting processes. The manuscript was revised to:**

**(1)** **Justify/contextualize the lab method we chose for estimating S (as a common ex situ estimation method that may overestimate natural S).**

**(2) Show that the estimates match those from other similar plants from other estimation methods.**

**(3) Explicitly discuss the differences between the lab estimates and natural wetting.**

3) I have additional reservations regarding the interpretation. In the results, 0.43 mm was introduced as an estimate of the specific stem storage capacity, but in the discussion this value is explained by (l 296) the "large densities, up to 700,000 stems ha-1 (Dias et al., 2018) – 56,770 stems ha-1 at our study site." I fail to see how the density has a direct effect on the specific storage.

**Response: We agree that the scaling of stem and canopy S could be clearer. As such, we have added an equation to the methods (Equation 2). This should clarify the two specific points raised by the reviewer:**

**(1) The units for specific S should have been mL cm-2, thus the specific stem S of 0.436 mm is really 0.436 mL cm-2. According to equation 2, scaled stem S turns out to be a similar number as specific stem S, but with different units:**
**Scaled stem S = 0.425 [L/m2] = 0.436 [mL/cm2] * 171.9 [cm2/plant] * 5.68 [plants/m2] / 1000**

**(2) Regarding the direct effect of density on scaled stem S, this is now clarified per equation 2 which shows that the number of stems ha-1 is used in scaling specific stem S.**

[revised manuscript text omitted]